# Assessment of Carbon Emissions at the Logistics and Transportation Stage of Prefabricated Buildings

Yichen Zhang [1], Tian Peng [2], Chao Yuan [1,*] and Yang Ping [3]

1  School of Civil Engineering, Shandong University, Jinan 250012, China
2  The Second Construction Company Ltd., China Construction Second Bureau, Shenzhen 518000, China
3  China State Construction Second Engineering Bureau Co., Ltd., Beijing 100161, China
*  Correspondence: chao.yuan@sdu.edu.cn

**Abstract:** As regards the carbon emission levels of the logistics stage of prefabricated buildings, this study aims to fill the gap in scientific and unified carbon emission calculation models and standards. Thus, the calculation boundary for carbon emissions was first defined in this study. Secondly, various carbon emission factors related to China's energy consumption characteristics were summarized. Subsequently, a carbon emission calculation model for the logistics stage was established, based on the carbon emission factor method. Finally, taking a project as an example, the carbon emission level at the transportation stage was calculated using the proposed model. The effect of full-load rates on the carbon emission levels of prefabricated components was also evaluated. The results demonstrate that the full-load rate has a significant effect on carbon emissions during the transportation stage (54.32% reduction in carbon emissions at 100% full load). Therefore, increasing the full-load rate can reduce carbon emissions from transportation, as long as the loading requirements are met.

**Keywords:** carbon emission calculation; industrial prefabricated building; logistics and transportation stage; full-load rate

## 1. Introduction

Massive emissions of greenhouse gases have led to global warming. This has caused extreme climate disasters, such as glacier retreat, melting permafrost, droughts, and torrential rains. These events pose a great threat to the natural balance of ecosystems and human survival. As the most significant of greenhouse gases, carbon dioxide ($CO_2$) emissions have become an important indicator of total greenhouse gas emissions. The 26th United Nations Climate Change Conference (COP26) summit of the United Nations Framework Convention on Climate Change (UNFCCC) in 2021 urged parties to accelerate emission reduction actions and to align national climate action commitments with the Paris Agreement. This was designed to keep alive the hope of limiting global temperature rise to 1.5 °C and setting a net zero emission goal for 2050 [1]. The impact of this conference on the construction industry is, therefore, immediate and far-reaching.

The construction sector accounts for approximately 42% of global greenhouse gas emissions [2], with $CO_2$ accounting for 82.9% of greenhouse gas emissions. As a major construction country, China is committed to reducing emissions and controlling temperatures to mitigate climate change, in line with the trend of the global climate governance process. China has adopted carbon emission reduction, carbon peaking, and carbon neutrality as its national strategy, aiming to achieve carbon peaking by 2030 and carbon neutrality by 2060. Therefore, reducing the energy consumption and carbon emissions in the construction industry has become an unstoppable trend.

The prefabricated construction mode is production–transportation–assembly, in which the components of a building are produced in a factory and then transported to the construction site for assembly. Pons et al. used the life cycle assessment (LCA) method to

analyze 200 prefabricated schools, demonstrating that prefabricated technology was more effective in reducing carbon emissions and was more environmentally friendly [3]. Therefore, advocating for the development of prefabricated buildings could help the construction industry become more industrialized. In addition, it could lead to energy use reduction over the entire life cycle of a building project. This is necessary in order to change and improve the Chinese construction industry.

The entire life cycle of assembled buildings includes the materialization stage, operation stage, and demolition stage. Among these, the materialization stage is the stage with the largest carbon emissions [4]. This can be divided into three stages: factory production, logistics and transportation, and on-site assembly. By comparing the carbon emission levels of the prefabricated construction mode to those of the traditional construction methods, it can be seen that the proportion of carbon emissions in the production and assembly construction stages of assembled buildings is lower than that of traditional cast-in-place buildings. In contrast, the proportion of carbon emissions in the logistics and transportation stages is higher for the assembled buildings [5]. Regarding the prefabricated assembly building mode, currently, the transported items are no longer scattered building raw materials, but are instead prefabricated components. Given the characteristics of prefabricated components, which include large size and quantity, complex stacking, and heavy weight, it is frequently necessary to transport them in multiple vehicles. This is the main reason for the increase in carbon emissions in the transportation phase of prefabricated buildings.

The logistics and transportation industry has one of the largest carbon emission rates, and the $CO_2$ generated by the logistics industry in China currently accounts for more than 25% of the total $CO_2$ emissions [6]. Furthermore, it will take a long time for new energy vehicles to replace fossil fuel vehicles; therefore, it is of far-reaching significance to explore the transportation of prefabricated buildings.

Few studies, however, have been conducted on the calculation of carbon emissions during the transport phase of prefabricated buildings in China. In these studies, the calculation boundaries were generally vague. For example, studies which relate to the carbon emission levels of transportation methods, such as Johnson's comparison of the carbon emission levels of two types of forklifts (electric and LPG) [7] and Liu, Fuming et al.'s calculation/comparison of the energy consumption/carbon emissions of truck transportation and belt conveyors through a carbon emission calculation model [8], have laid the foundation for the calculation and optimization of the inherent carbon emissions in the transportation of prefabricated buildings. Simultaneously, they have provided research ideas and methods.

This study, therefore, addresses the shortcomings of the existing literature through evaluation of the transportation phase of the logistics of assembled buildings, based on the existing building carbon emission formulas. Furthermore, this study establishes a calculation model for the transportation phase of assembled buildings through a clear delineation of the calculation boundary. In addition, considering the particularity of the transportation of prefabricated components, it was necessary to consider the factor of the full-load rate in the transportation of prefabricated components. This ensured that the carbon emission calculation was more consistent with the actual situation. Subsequently, the proposed calculation model was used to determine the carbon emission levels. The effect of the full-load rate on carbon emissions during the transportation of prefabricated and assembled buildings was also examined.

The remainder of this paper is organized as follows. In Section 2, the research boundaries and key parameters are identified. A calculation model for the carbon emissions of prefabricated buildings in the logistics and transportation stage (considering the influence of the full-load rate) is also proposed. The study results are provided in Section 3. The influence of different full-load ratios on the transportation stage of prefabricated components, through case analysis, is discussed in Section 4. Finally, conclusions are drawn in Section 5.

## 2. Research Boundary and Key Parameters

### 2.1. Calculation Boundary of Carbon Emissions [9]

Carbon emissions are greenhouse gas emissions caused by the entire process of extracting, producing, and using energy [10]. The main greenhouse gases are methane ($CH_4$), nitrous oxide ($N_2O$), and hydrofluorocarbons (HFCs). In this study, carbon dioxide ($CO_2$) equivalent was used as the calculation boundary for greenhouse gases used for carbon emission calculations.

In this study, the transportation phase of the assembled building, which includes the transportation of general building materials and prefabricated components, was examined. The study boundary of building material transportation is the process of transporting building materials from the production site to the construction site. The inherent carbon emissions include direct carbon emissions from the transportation process and indirect carbon emissions from the production process of the consumed energy [11]. The study boundaries of prefabricated component transportation include the loading of transportation vehicles at the factory, transportation from the factory to the construction site, and unloading at the construction site. In this phase, we do not consider the environmental impact associated with moving large pieces of equipment into and out of the plant on the construction site or the secondary transportation that comes from construction work.

### 2.2. Carbon Emission Measurement Methods

Currently, the commonly used carbon emission measurement methods can be roughly divided into the following three types: the emission factor method [12], the mass balance method [13], and the actual measurement method [14] (see Table 1). The emission factor is the amount of carbon dioxide emitted per unit of product produced or energy consumed [15]. This method calculates carbon emissions by taking the average amount of $CO_2$ released and turning it into a carbon emission factor. This is usually available as a default value from the Intergovernmental Panel on Climate Change (IPCC). The formula for calculating carbon emissions is shown in Equation (1):

$$\text{Carbon emissions} = \text{Carbon emission factor} \times \sum \text{activity data.} \tag{1}$$

**Table 1.** Comparison of carbon emission measurement methods [13–15].

| Category | Advantages | Disadvantages | Applicable Scale |
|---|---|---|---|
| **Emissions Factor method** | 1. Easy to calculate 2. Authoritative database of carbon emission factors is available 3. Not limited by testing instruments | 1. Poor accuracy 2. Large differences in carbon emission factor database | Macroscopic Microscopic |
| **Real Measurement method** | 1. High precision | 1. High cost 2. Data not easily available | Microscopic |
| **Mass Equilibrium method** | 1. Higher precision 2. Carbon emission sources are clear | 1. High workload 2. Data may not be representative | Macroscopic |

Greenhouse gas emissions and energy consumption are measured near the carbon emission sources using professional testing equipment.

Based on the analysis method of the law of conservation of mass, a model is made to statistically examine the substances that go into the research object and where they go when it is done.

In summary, for this study, we adopted the emission factor method as the carbon emission measurement method for the transportation stage of assembled prefabricated

buildings. Carbon emissions are a product of the energy consumption of a certain transportation means and the carbon emission factor of the energy consumed.

*2.3. Selection of Carbon Emission Factors*

The carbon emission factors involved in the entire life cycle of prefabricated buildings mainly include energy carbon emission factors, construction material carbon emission factors, transportation tool carbon emission factors, and machinery and equipment carbon emission factors. In contrast, the logistics and transportation stage mainly involves three carbon emission sources: energy, transportation tools, and machinery and equipment.

Carbon emissions associated with energy in the logistics and transportation stage can be roughly divided into fossil energy carbon emissions and electricity carbon emissions. Electricity energy does not produce carbon emissions during use but emits $CO_2$ during production [16], hence the need to consider the impact of electricity on carbon emission levels in this study. The logistics transportation phase is mainly carried out by rail, road, and waterway; therefore, only the carbon emission factors generated by the means of transportation involved in these three transportation modes were considered in this study. Carbon emissions from mechanical equipment generated during the vertical transportation (loading and unloading) of prefabricated components also need to be considered—mainly the consumption of gasoline, diesel, and electricity (where energy is mainly consumed by the gantry crane in the component plant and the tower crane at the construction site) [17]. The differences in fuel types, extraction and processing technologies, and testing methods have led to different values of energy carbon emission factors measured by different organizations. To ensure the authenticity and reliability of the data, we gave priority to the data applicable to China's national conditions. In order to summarize and calculate the prefabricated assembled fossil fuel carbon emission factors for the building phase, the carbon content per unit calorific value and carbon oxidation rate, recommended by the 2006 IPCC National Greenhouse Gas Inventory and the Carbon Emission Calculation Standard for Buildings (GB/T 51366-2019), were utilized. The average low-level heat generation value was sourced from the 2020 China Energy Statistical Yearbook. To facilitate the calculation, fossil fuel carbon emission factors were expressed firstly, in different units of measurement, as shown in Table 2. The units were then converted to common domestic units using Equation (3) ($kgCO_2$/kg). The discount standard coal coefficient is the average low calorific value of fuel divided by the low calorific value of standard coal, when using standard coal as a unit. Raw coal can be divided into three types based on the degree of carbonization: lignite, bituminous coal, and anthracite. The average low-level heat generation value was used to determine the heat generation value of raw coal.

$$\mathrm{DEF} = \mathrm{DCC} \times \mathrm{DOF} \times \frac{44}{12}; \qquad (2)$$

$$\mathrm{CEF} = \mathrm{DEF} \times \mathrm{Avg.LHV}. \qquad (3)$$

DEF: Default emission factor.
DCC: Default carbon content.
DOF: Default oxidation factor.
Avg. LHV: Average low-level heat generation value.
CEF: Carbon emission factor.

**Table 2.** Summary of $CO_2$ emission factors for various fossil fuels.

| Type | Energy Source Name | Average Low-Level Heat Generation (KJ/kg) | Discounted Standard Coal Coefficient (kgce/kg) | Unit Calorific Value Carbon Content (tC/TJ) | Carbon Oxidation Rate (%) | Emission Factor (tCO₂/TJ) | Emission Factor (kgCO₂/kg) | Emission Factor (kgCO₂/kgce) |
|---|---|---|---|---|---|---|---|---|
| **Solids** | Anthracite | 20,908 | 0.7143 | 27.4 | 0.94 | 94.44 | 1.97 | 2.76 |
| | Bituminous coal | 20,908 | 0.7143 | 26.1 | 0.93 | 89.00 | 1.86 | 2.61 |
| | Lignite | 20,908 | 0.7143 | 28.0 | 0.96 | 98.56 | 2.06 | 2.88 |
| | Coke | 28,435 | 0.9714 | 29.5 | 0.93 | 100.60 | 2.86 | 2.94 |
| | Type coal | 20,908 | 0.7143 | 33.6 | 0.90 | 110.88 | 2.32 | 3.25 |
| **Liquids** | Crude oil | 41,816 | 1.4286 | 20.1 | 0.98 | 72.23 | 3.02 | 2.11 |
| | Gasoline | 43,070 | 1.4714 | 18.9 | 0.98 | 67.91 | 2.93 | 1.99 |
| | Diesel | 42,652 | 1.4571 | 20.2 | 0.98 | 72.59 | 3.10 | 2.12 |
| | Fuel oil | 41,816 | 1.4286 | 21.1 | 0.98 | 75.82 | 3.17 | 2.22 |
| | Kerosene | 43,070 | 1.4714 | 19.6 | 0.98 | 70.43 | 3.03 | 2.06 |
| | Liquefied LPG | 50,179 | 1.7143 | 17.2 | 0.98 | 61.81 | 3.10 | 1.81 |
| **Gas** | Natural gas | 38,931 | 1.3300 | 15.3 | 0.99 | 55.54 | 2.16 | 1.63 |

There are significant regional differences in the carbon emissions associated with power, with northeastern and northern China power grids dominated by thermal power generation, which has a large carbon emission factor. Central and southern China are dominated by hydro power generation; therefore, their carbon emission factor is smaller. Based on the data provided by the IPCC and the research results of references [18,19], the carbon emission factors of power energy for different regions of China are summarized in Table 3.

**Table 3.** Summary of carbon emission factors of electric energy.

| Region | Carbon Emission Factor (kgCO₂/kWh) | Region | Carbon Emission Factor (kgCO₂/kWh) |
|---|---|---|---|
| East China Power Grid | 1.04 | Central China Power Grid | 0.848 |
| North China Power Grid | 1.27 | Northwest China Power Grid | 0.944 |
| Northeast Power Grid | 1.36 | Southern Power Grid | 0.854 |
| National average | | 1.05 | |

Considering the types, weights, volumes, stacked layers, and transportation costs of building materials and prefabricated components, vehicles with different loads and power types need to be selected for transportation. The Standard for Calculating Carbon Emissions of Construction (GB/T 51366-2019) factors for carbon emissions from transportation are listed in Table 4.

**Table 4.** Summary of carbon emission factors for each type of transportation.

| Transportation Mode | Type of Transportation Vehicle | Rated Load Capacity (t) | Fuel Type | Carbon Emission Factor [kgCO₂ₑ/(t·km)] |
|---|---|---|---|---|
| Road | Light truck | 2 t | Gasoline | 0.334 |
| | | 2 t | Diesel | 0.286 |
| | Medium-sized truck | 8 t | Gasoline | 0.115 |
| | | 8 t | Diesel | 0.179 |
| | Heavy truck | 10 t | Gasoline | 0.104 |
| | | 18 t | Gasoline | 0.104 |
| | | 10 t | Diesel | 0.162 |
| | | 18 t | Diesel | 0.129 |
| | | 30 t | Diesel | 0.078 |
| | | 46 t | Diesel | 0.057 |

**Table 4.** *Cont.*

| Transportation Mode | Type of Transportation Vehicle | Rated Load Capacity (t) | Fuel Type | Carbon Emission Factor [kgCO$_{2e}$/(t·km)] |
|---|---|---|---|---|
| Railroad | Internal combustion locomotive | | Diesel | 0.011 |
| | Electric locomotive | | Electric power | 0.010 |
| Waterway | Liquid cargo ship | 2000 t | Fuel oil | 0.019 |
| | Dry bulk carrier | 2500 t | Fuel oil | 0.015 |
| | Container ship transportation | 200 TEU | Fuel oil | 0.012 |

t = ton; TEU = twenty-foot equivalent unit.

Based on the energy carbon emission factors collated above, and the Rules for the Preparation of Construction Machinery Shift Costs for Construction Projects, which can be used to obtain the energy consumption per unit shift of common logistics transportation equipment, the carbon emission factors of machinery and equipment were calculated using Equation (4) and are shown in Table 5.

$$\text{CEF of machinery and equipment = EC pu shift of each machinery and equipment} \times \text{EC pu shift} \qquad (4)$$

CEF of machinery and equipment: Carbon emission factor of machinery and equipment.
EC pu shift of each machinery and equipment: Energy consumption per unit shift of each machinery and equipment.
EC pu shift: Energy consumption per unit shift.

**Table 5.** Summary of carbon emission factors of machinery and equipment commonly used in the transportation stage of prefabricated construction.

| Name of Machinery | Model Specification | Energy Type | Energy Consumption per Shift (kg) | Carbon Emission Factor (kgCO$_{2e}$/shifts) |
|---|---|---|---|---|
| Self-lifting tower crane | Lifting capacity 400 t | Electricity | 164.31 | 172.53 |
| | Lifting capacity 60 t | Electricity | 166.29 | 174.60 |
| | Lifting capacity 800 t | Electricity | 169.16 | 177.62 |
| | Lifting capacity 1000 t | Electricity | 170.02 | 178.52 |
| | Lifting capacity 2500 t | Electricity | 266.04 | 279.34 |
| | Lifting capacity 3000 t | Electricity | 295.60 | 310.38 |
| Portal-type crane | Lifting capacity 10 t | Electricity | 88.29 | 92.70 |
| Dump truck | Load capacity 5 t | Gasoline | 31.34 | 91.83 |
| | Load capacity 15 t | Diesel | 52.93 | 164.08 |
| Truck with load | Load capacity 4 t | Gasoline | 25.48 | 74.66 |
| | Load capacity 6 t | Diesel | 33.24 | 103.04 |
| | Capacity 8 t | Diesel | 35.49 | 110.02 |
| | Capacity 12 t | Diesel | 46.27 | 143.44 |
| | Load capacity 15 t | Diesel | 56.74 | 175.89 |
| | Capacity 20 t | Diesel | 62.56 | 193.94 |
| Flatbed trailer | Capacity 20 t | Diesel | 45.39 | 140.71 |

The national average value was chosen as the carbon emission factor of electric energy in Table 5.

## 3. Calculation Modeling

Unlike the transportation stage of the traditional construction method, the transportation stage of an industrialized prefabricated assembly building contains two parts: transportation of building materials and transportation of prefabricated components. The total carbon emission formula for this stage is shown in Equation (5). In this study, we consider the transportation of building materials and the transportation of prefabricated components as the research objects. Thus, we establish a carbon emission calculation

formula, based on the above-mentioned carbon emission factors, to reflect the real carbon emissions of the logistics and transportation stages relatively accurately.

$$E_t = E_{tp} + E_{tu} \tag{5}$$

$E_t$: Total carbon emissions during the logistics stage.
$E_{tp}$: Carbon emissions from the logistics phase of prefabricated components.
$E_{tu}$: Carbon emissions from the logistics phase of building materials.

### 3.1. Calculation of Carbon Emissions from the Transportation of Building Materials

The transportation phase of building material logistics is the transportation of materials from the factory to the construction site, without considering the carbon emissions generated by the transportation of raw materials from the mining site to the material processing plant. Differences in carbon emissions caused by road conditions and transporters' operation of the means of transport were not considered. The transportation of building materials to a construction site leads to greenhouse gas emissions. The carbon emissions from the transportation of building materials were calculated using the distance method, which is mainly related to the transportation distance, the weight and type of materials, the transportation method, and the transportation time [20]. Its calculation formula is as follows:

$$E_{tu} = \sum_{i=1}^{m} \left( \sum_{i=1}^{n} Q_{ci} \times D_{ci,j} \right) \times f_{Tj} \tag{6}$$

$Q_{Ci}$: Demand for building material $i$.
$D_{ci,j}$: Distance that building material $i$ is transported by transport mode $j$.
$i$: Building material type.
$j$: Mode of transport of building materials.
$\sum_{i=1}^{n} Q_{ci} \times D_{ci,j}$: Summation of distances of all construction materials transported using the transport method $j$.
$f_{Tj}$: Carbon emissions factor of transportation mode $j$.

### 3.2. Carbon Emission Calculation of Precast Component Transportation

Prefabricated component transportation includes two carbon emission sources, off-site transportation and on-site transportation, which can be subdivided into three parts: primary vertical transportation, horizontal transportation, and secondary vertical transportation. The carbon emission calculation considers the carbon emissions of the transportation vehicles used for transportation and the energy consumption of the mechanical equipment used for loading and unloading. The calculation process is as follows:

$$E_{tp} = E_{tp1} + E_{tp2} + E_{tp3} \tag{7}$$

$E_{tp1}$: Carbon emissions from the precast component loading stage.
$E_{tp2}$: Carbon emissions from the transportation phase of the precast components.
$E_{tp3}$: Carbon emissions from the unloading stage of precast components.

$$E_{tp1} = E_{tp3} = \sum_{q}^{n} \sum_{z=1}^{n} r_{qz} p_{qz} \tag{8}$$

$q$: The type of energy consumed by type z machinery per shift.
$r_z$: The carbon emission factor for machinery of type z.
$p_z$: The number of machine shifts of type z.

The transportation of prefabricated components cannot reach a full load because of the different sizes of prefabricated components and the fact that too many stacked layers can affect structural safety and building life, resulting in the carbon emission factor of the transportation phase being influenced by the full-load rate [21]. In contrast to the general

transportation of building materials at full load, the effect of the full-load rate must be considered when calculating the transport of prefabricated components. This means that the carbon emission factors of the means of transport at different full-load rates need to be considered.

$$E_{\text{tp2}} = \sum_{j=1}^{b2} \left( \sum_{k=1}^{u} D_{Pk,j}^2 \times Q_{Pk} \right) \times f_{Tj1z} \tag{9}$$

$j$: Mode of transportation of prefabricated components.

$k$: Type of prefabricated component.

$Q_{Pk}$: Demand for prefabricated component $k$.

$D_{Pk,j}^2$: Transportation distance of the prefabricated component $k$ by transport mode $j$.

$\sum_{k=1}^{u} D_{Pk,j}^2 \times Q_{Pk}$: The sum of the distances to the site for all components using transport mode $j$.

$f_{Ti1z}$: Carbon emission factor for transport mode $j$ when considering the full-load factor.

## 4. The Case Study of Jinan Building Construction Site

We consider a project in the Jinan area in China as an example. The project included five simply furnished 12-story public rental housing residential buildings with a total construction area of 20,327.1 m², all built via prefabricated assembly construction. Building 2 of the project was selected for the carbon emission analysis, with a GFA of 3876.1 m². The total steel consumption of the project was 31,003.27 kg, the concrete consumption was 1810.01 m³, and the wood consumption was 27,802.27 kg. The average transportation distance of building materials was obtained from transportation companies and building material suppliers. Building material transportation vehicles were all heavy-duty diesel trucks, with a rated load capacity of 30 T. According to the Standard for Calculation of Construction Carbon Emissions (GB/T 51366-2019), it can be determined that the carbon emission factor per tonne of building materials transported in the starting logistics phase is 0.078 $kgCO_{2e}/(t\Delta km)$. The carbon emissions of the transportation process of building material logistics are summarized in Table 6 where 1 m³ C30 concrete weighs approximately 2400 kg.

**Table 6.** Carbon emissions from the transportation of major building materials.

| Name of Building Material | Unit | Quantity | Average Transportation Distance (kg) | Carbon Emission Factor [kgCO₂ₑ/(t·km)] | Total Carbon Emissions (kg) |
|---|---|---|---|---|---|
| Steel | kg | 310,003.27 | 130.89 | 0.078 | 3164.95 |
| Concrete C30 | m³ | 1810.01 | 68.22 | 0.078 | 23,115.25 |
| Wood | kg | 27,802.27 | 58.03 | 0.078 | 125.84 |
| Total | | | | | 26,406.04 |

The loading and unloading of prefabricated components mainly relies on gantry cranes and tower cranes, and electricity is the main energy consumed. Since Jinan is in East China, its electricity carbon emission factor is 1.04 $kgCO_2/kWh$. The carbon emissions of on-site transportation are shown in Table 7.

**Table 7.** On-site transportation carbon emissions.

| Energy | Total Energy Consumption | Unit | Carbon Emission Factor (kgCO₂ₑ/units) | Total Carbon Emissions (kg) |
|---|---|---|---|---|
| Electricity consumption (kWh/m²) | 11,020.89 | kWh | 1.04 | 11,461.73 |

For the prefabrication rate of the assembled building and the prefabricated component transportation vehicle, the Changan Star 9 compartment transporter was selected. The energy source was 92 gallons of gasoline, and its rated capacity was 630 kg. The total weight of transported prefabricated components for one building was 3271.08 t. Values for the transportation of prefabricated components and carbon emission factors of gasoline vehicles, under different full-load rates (Table 5), were utilized in carbon emission simulations to obtain carbon emission values under different full-load rates, as shown in Table 8. Based on actual engineering experience, the one-way distance of the assembly was taken as 100 km.

**Table 8.** Carbon emissions from the transportation of prefabricated components under the influence of different full-load rates.

| Transportation Volume (t) | Gasoline Vehicle | | | |
| --- | --- | --- | --- | --- |
| | Full-Load Rate (%) | Transportation Carbon Emission Factor (Considering Full-Load Rate) [$kgCO_{2e}/(t·km)$] | One-Way Transport Distance (km) | Carbon Emission (kg) |
| **3271.08** | 50 | 0.5694 | 100 | 186,255.30 |
| | 60 | 0.4434 | | 145,039.69 |
| | 70 | 0.3674 | | 120,179.48 |
| | 80 | 0.3181 | | 104,053.05 |
| | 90 | 0.2843 | | 92,996.80 |
| | 100 | 0.2601 | | 85,080.79 |

Figure 1 compares the carbon emission levels of each component of the logistics and transportation stage. In Figure 1, "a" represents the transportation of building materials, "b" represents the loading and unloading of prefabricated components, and "c" represents the off-site transportation of the prefabricated components. The off-site transportation of the prefabricated components was plotted at a full-load rate of 70%.

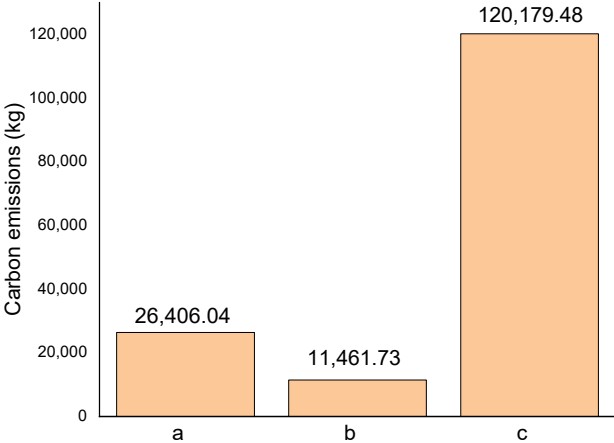

**Figure 1.** Comparison of carbon emissions of each part in the logistics and transportation stage.

Figure 2 shows the carbon emission calculation of precast transportation under different full-load rates.

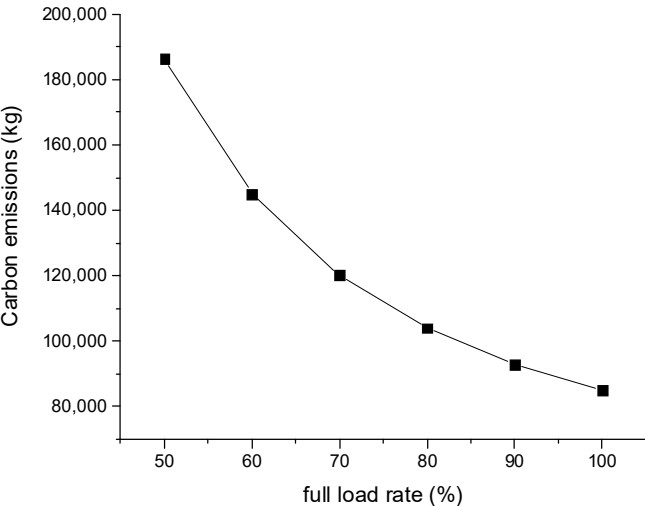

**Figure 2.** Carbon emission variation of prefabricated component transportation under different full-load rates.

## 5. Discussion

Figure 1 presents the carbon emissions of general building materials, precast loading and unloading, and off-site transportation of precast components, where the carbon emissions of off-site transportation of precast components correspond to a full-load rate of 70%. The graph shows that the carbon emission level of the transportation stage of prefabricated components accounts for a larger proportion of the carbon emissions of the transportation stage of assembled buildings, at 76%. The transportation of building materials and loading and unloading of prefabricated components account for smaller proportions, at 16.71% and 7.25%, respectively. As can be deduced from the carbon emissions proportions in the figure, it is clear that the accuracy of the carbon emission calculations for the whole transportation phase depends on how well the transportation of the prefabricated components is calculated.

In Figure 2, it can be seen that the carbon emission level of the precast components decreases with an increase in the full-load rate, showing a non-linear trend. Compared with a full-load rate of 50%, the carbon emission levels in the transportation phase of the assembled building could be reduced by 54.32% at a full load of 100%. The figure indicates that different full-load rates have a large effect on the carbon emission level of prefabricated components during the transportation phase. This demonstrates the importance of taking the full-load rate into account when determining the value of the carbon emission factor. It also proves that the formula used for calculating the carbon emission level during the transportation phase of prefabricated components was accurate.

## 6. Conclusions

The percentage level of carbon emissions in the production stage and on-site construction stage of assembled components will decrease with improvements in prefabricated assembly technology. In contrast, the percentage level of carbon emissions in the logistics and transportation stages will increase significantly; therefore, it is crucial to develop scientific and reasonable carbon emission calculation standards for logistics and transportation. This study addresses and clarifies the boundary gap and standard gap of the current transportation stage of assembly through the existing carbon emission calculation standards for buildings combined with the characteristics of assembly buildings. We propose the use of the emission factor method to calculate carbon emissions for the transportation of building materials and the transportation of prefabricated components, in order to improve the operability of carbon emissions in the transportation stage of prefabricated assembly buildings.

Meanwhile, the full-load rate factor, which will depend on requirements such as the size of the components, was considered in the transportation phase of the assembled components. The full-load rate factor was also considered in the calculation formula in order to improve the transportation of the prefabricated components.

Finally, we showed through a case study that the full-load rate has a large impact on the carbon emission level of precast component transportation and the calculation results of the entire transportation phase of assembled buildings. Thus, it should not be ignored in actual calculations. It is also important to note that data on the carbon emission factors under different full-load rates are still missing. It is, therefore, suggested that the effect of full-load rates be considered in the future development of carbon emission standards for assembled buildings. Finally, the industry default values should be adjusted to make the calculation results more accurate and objective.

**Author Contributions:** Conceptualization, C.Y., Y.Z. and T.P.; methodology, C.Y. and Y.Z.; software, Y.Z. and T.P.; validation, T.P., C.Y. and Y.P.; formal analysis, C.Y. and Y.P; investigation, T.P.; resources, Y.P.; data curation, Y.Z. and T.P.; writing—original draft preparation, Y.Z. and Y.P.; writing—review and editing, Y.Z. and T.P.; visualization, Y.P. and T.P.; supervision, C.Y.; project administration, C.Y., Y.P. and T.P.; funding acquisition, C.Y., Y.P. and T.P. All authors have read and agreed to the published version of the manuscript.

**Funding:** This research was funded by a research project (Grant No. 2021ZX000002 and Grant KJYFKT2022002) of China State Construction Second Engineering Bureau Co., Ltd. The research was also supported by the National Natural Science Foundation of China (Grant No. 52108373) and the Natural Science Foundation of Shandong Province (Grant No. ZR2021QE127).

**Institutional Review Board Statement:** Not applicable.

**Informed Consent Statement:** Not applicable.

**Data Availability Statement:** Not applicable.

**Conflicts of Interest:** Author Tian Peng was employed by the company The second Construction Company Ltd of China Construction Second Bureau. And author Yang Ping was employed by the company China State Construction Second Engineering Bureau Co. Ltd. The remaining authors declare that the research was conducted in the absence of any commercial or financial relationships that could be construed as a potential conflict of interest.The authors declare that this study received funding from China State Construction Second Engineering Bureau Co.Ltd. The funder was not involved in the study design, collection, analysis, interpretation of data, the writing of this article or the decision to submit it for publication.

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
