# Peer review of "Assessment of Carbon Emissions at the Logistics and Transportation Stage of Prefabricated Buildings"

_applsci, doi:10.3390/app13010552_

Round 1
Reviewer 1 Report
The paper presents a computational analysis on the carbon emissions associated with the logistics of prefabricated buildings in China, including loading, transportation and unloading of precast building products. While the context and the results of the study may be of interest to some practitioners and industry audience, the methodology is nothing novel and makes no theoretical contributions. Contributions to new knowledge are also very limited beyond just a case study of practical project work.
In addition, there appear some errors/issues in the paper as follows:
1) In Table 2, the same metrics used for the Emission Factor in the last two columns, i.e. kgCO2/kg, whereas the values presented are different.
2) In Eq.4, Dci,j is for the distance that building material i is transported by mode j. However, it is unclear whether this considers accumulated distance when the material i involves multiple deliveries using mode j.
3) In Eq.7, the factor f (i.e. the carbon emission factor for transport j when the full load is applied) appear missing from the equation.
4) In Table 9, it is questionable why only consider "one-way transport distance" for all load fractions, since multiple trips (i.e. running back and forth) would be needed to deliver the required quantity.
5) In the last paragraph of Section 4, it states that "From Figure 2, it can be seen that the carbon emission of precast components increases with the increase of the full load rate". In fact, the figure shows the emission decreases when the full load rate increases. Also, the percentage of reduction in carbon emissions from 50% full load rate to 100% full load rate is the same as that of the difference between their respective transportation carbon emission factors (as shown in Table 9). So, Figure 2 does not add any new information.
Overall, the paper is not suitable to be published as a research article.
Author Response
Dear Reviewer,
We appreciate the comments of you. The responses that are included in the revised paper are documented in the following. Revisions within the paper are as follows: The added content is highlighted in yellow, and the deleted content is marked with red deletion line.
The writing has been revised throughout and we have gone over the manuscript in detail. We believe that all the proposed suggestions have been followed and satisfied.
We hope for a positive response.
Sincerely,
Chao Yuan
Shandong University

Reviewer 2 Report
The paper tackles the problem of carbon emission estimation of the transport and logistics phase of building construction, with a case study and data for the situation in China, based on the emisions data for China.
The article explains the applied methodology, and why it was chosne compared to the other alternatives.
The article is generally well structured, although it could benefit from slight improvement of the english, avoiding some general phrases at the begining of some statements etc. Some references could be added to make a stronger stand point of the article. One of souch examples is tha statement in row 26, stating that China has always takan care about the carbon emision reduction. Since CO2 emision reduction is a relatively new trend, the authors should find refences to prove souch a claim, or change the statement by putting it in a relevant temporal context also including references (souch as official national stretegic documents, action plans, legal documents etc.). In row 57, it is explained that China has not developed an official method for calculating CO2 emisions during the proces of production of prefabricated buildings. This also seems as an overstatement, and should be linked to a reference, stating that China has started a process of designing souch a specific official method, which is not a common practice in most countries. Otherwise, it would be sufficient to say that the article tackles this specific problem and applies a specific method to do so.
The entire text should be checked for technical and language errors, typos etc. (e.g. the term "boundary" is used many times in the text, but there are many better terms in english which could be used to beter transfer the desired message to a reader; in row 72 a therm "whenamounts" shoudl be corrected and so on...)
THe authors should explain the aberivation IPCC the frist time it appears in the text.
The equation in row 93, should be written as an equation, not as part of the text.
From the paragraph starting in row 114, one could assume that the electricity consumption and its effect on CO2 emissions would be neglected in the study, but they are accounted for. The authors should rewrite this to avoid confusion.
Row 234 "4. Case study", shoudl be changed to "The case study of Jinan building construciton site" or similar.
The case study results should be better presented. The authours could attempt to use more diagrams and graphs to better stress out the difference in carbon emisions for prefabricated and standard on-site contraction. Furthermore, the results and conclusion coudl benefit from an analysis of critical factors which could have an effect on the results, since each construciton site has a different location and different conditions.
Furhtermore, the research would singificantly benefit from including an analysis of the CO2 emsions from during the construction material production phase, including the production of prefabricated elements. This way, a more realistic comparison of these 2 alternatives could be performed, and the conclusions would be better and more applicable.
Author Response

(The authors gave the same response as above.)

Reviewer 3 Report
The manuscript titled “Research on carbon emissions in the logistics and transportation stage of prefabricated buildings” has been reviewed. Here are my minor suggestions which can improve the manuscript.
First of all, the idea of the analysis is very relevant. In fact, given the urgency of climate change, all research focused on the economic events that could harm the environment are vital. Briefly, I have a good time reading this paper, but some revisions are needed before being published.
Here are my comments:
• Overall, I recommend the inclusion of an acronyms table before the introduction section. In my opinion, there is a lot of acronyms, which could make it difficult to the readers to have it present along all paper. So, a table might help.
• The introduction part is required to add few more sentences to increase the strength of this article and kindly bring in the research problem, objective, novelty and explain it in last paragraph of the section of Introduction.
• Prior to this sentence “The construction sector accounts for about 42% of global greenhouse gas emissions [1], with CO2 emissions accounting for 82.9% of greenhouse gas emissions” I will suggest you give general information on environmental degradation and the contribution of CO2 emissions towards it
• Information on the recent COP26 is missing in the introduction section. Kindly incorporate such information since it is relevant to your study
• The discussion section needs minor improvement. This can be done by comparing your results with prior studies.
• Use of English language need slight improvement. Revise almost all sentences in the manuscript with appropriate use of grammar, punctuation and speech.
Author Response

(The authors gave the same response as above.)

Reviewer 4 Report
The scientific problem of the article is clearly revealed. However, when presenting a scientific problem, a broader analysis of scientific articles by foreign authors is missed.
The aim and tasks of the paper should be defined more detail and clear.
106. Table 1: The article presents three carbon emission measurement methods. However, there is a lack of references to the literature sources on which these three methods are presented and discussed.
84. Chapter 2.2.1 „Emission factor method“ should not be divided into three more subsections, subsections 2.2.1, 2.2.2 and 2.2.3 should be combined.
143. The data source in Table 2. “Summary of CO2 emission factors for various fossil fuels” should be presented clearer.
1, 2 formulas: Formulas are usually written using abbreviations rather than writing the full names of the parameters used.
271 The description of the submitted Figure 1 must appear at the top of the figure.
273 „In the figure, In the figure,“ – should be corrected
278 The description of the submitted Figure 2 must appear at the top of the figure.
Author Response

(The authors gave the same response as above.)

Round 2
Reviewer 1 Report
The revised manuscript has addressed some of the comments listed in the previous review and appears improved in presentation and clarity.